# Development and Evaluation of a Slip Detection Algorithm for Walking on Level and Inclined Ice Surfaces

**DOI:** 10.3390/s22062370

**Published:** 2022-03-18

**Authors:** Jun-Yu Cen, Tilak Dutta

**Affiliations:** 1Institute of Biomedical Engineering, University of Toronto, Toronto, ON M5S 3G9, Canada; 2KITE Research Institute, Toronto Rehabilitation Institute-University Health Network, Toronto, ON M5G 2A2, Canada

**Keywords:** fall prevention, footwear, slips, slip detection, stride segmentation, inclined surface, winter, ice, machine learning, slip classification

## Abstract

Slip-resistant footwear can prevent fall-related injuries on icy surfaces. Winter footwear slip resistance can be measured by the Maximum Achievable Angle (MAA) test, which measures the steepest ice-covered incline that participants can walk up and down without experiencing a slip. However, the MAA test requires the use of a human observer to detect slips, which increases the variability of the test. The objective of this study was to develop and evaluate an automated slip detection algorithm for walking on level and inclined ice surfaces to be used with the MAA test to replace the need for human observers. Kinematic data were collected from nine healthy young adults walking up and down on ice surfaces in a range from 0° to 12° using an optical motion capture system. Our algorithm segmented these data into steps and extracted features as inputs to two linear support vector machine classifiers. The two classifiers were trained, optimized, and validated to classify toe slips and heel slips, respectively. A total of approximately 11,000 steps from 9 healthy participants were collected, which included approximately 4700 slips. Our algorithm was able to detect slips with an overall F_1_ score of 90.1%. In addition, the algorithm was able to accurately classify backward toe slips, forward toe slips, backward heel slips, and forward heel slips with F_1_ scores of 97.3%, 54.5%, 80.9%, and 86.5%, respectively.

## 1. Introduction

Falls are consistently among the most common causes of workplace injury [1]. They account for approximately 20% of the cost of all lost time injuries [2]. In 2019, US businesses spent over $10 billion on worker compensation claims for fall-related injuries and these costs are increasing [3]. During the winter, ice-covered walkways and stairs have been shown to increase the risk of slip and fall accidents [4,5]. According to the Canadian Institute for Health Information, falls on ice are the second most common cause of serious traumatic injuries, after motor vehicle collisions [6]. Although same-level falls on ice rarely cause immediate life-threatening injuries, they often result in injuries to the head, back, or lower extremities [7]. These falls can have particularly damaging effects on older adults either due to an injury triggering a downward spiral in health or from inactivity resulting from a fear of falling [8].

### 1.1. Slip-Resistant Footwear

Slip-resistant footwear can prevent falls on icy surfaces [9,10,11,12,13]. However, assessing the slip resistance on icy surfaces remains challenging. The current industry standard for measuring footwear slip resistance is a mechanical test developed by the SATRA Technology Centre and is recognized by ASTM International [14]. This method measures the coefficient of friction by applying a specified normal force pressing the test footwear onto a test surface and then moving the test surface horizontally at a set constant speed using the SATRA STM 603 Slip Resistance Testing machine. Unfortunately, this test has been found to have poor ecological validity [15,16,17] likely because it is unable to simulate the complex dynamics of human walking. Ecological validity for a test refers to whether the test results are representative of performance in real-life settings [18]. Recently, our team at the KITE Research Institute (Toronto Rehabilitation Institute, University Health Network) developed the human-centered Maximum Achievable Angle (MAA) test, which involves participants walking up and down ice-covered inclines while wearing test footwear in a simulated winter environment [15,16]. The slip resistance of the footwear is defined by the maximum slope angle (measured in degrees) that the participant can walk up and down without slipping [15]. Our testing has shown that the MAA test is able to identify differences in slip resistance performance that the SATRA test is unable to find [19]. In fact, our findings have shown that slip resistance varies widely among commercially available winter footwear and that a new generation of slip resistant winter footwear that incorporates composite materials performs much better than most other footwear available on the market [20,21]. These advance materials have also been shown to perform better than conventional footwear in real-world conditions [13]. A field study with 110 home healthcare workers found that the group wearing footwear that performed well on the MAA test reported nearly 80% fewer falls compared to a matched group wearing their own footwear [13]. The MAA test has also been used to evaluate how quickly these advanced composite materials lose their slip resistance performance, highlighting a potential limitation of this technology [22].

These findings are driving greater interest in the MAA test. However, there is one major limitation to this test method, which is that it relies on the subjective assessment of a human observer to determine when a participant has experienced a slip. These observers likely introduce variability in the results based on differences in vigilance and/or skill. We also suspect that MAA test observers disproportionally identify larger slips because these are easier to see. The addition of an automated slip detection system would make the MAA test more objective and accurate.

### 1.2. Existing Automated Slip Detection Methods

A number of methods for detecting slips have been described in the literature. Several studies have focused on the use of inertial measurement units (IMUs) to detect heel slips occurring on a straight and level path, neglecting toe slips that occur at toe off [23,24,25]. An IMU is a portable electronic device that measures and reports a body’s specific force, angular rate, and magnetic field using a combination of accelerometer, gyroscope, and magnetometer. Trkov et al. developed an algorithm using a set of five IMUs attached to the lower limb [25]. It detected slips based on how fast the shank pivoted after heel contact. It had a fast slip detection time of 59 ms, but no numerical accuracy in detection of slips vs. non-slips was reported. Hirvonen et al. used a waist-worn IMU to detect sudden movement caused by a person’s effort to regain balance [23]. However, it was only able to detect slips with slip distance greater than 5 cm. Lincoln et al. developed an insole sensor system with 90% slip detection accuracy using simple acceleration and force thresholds [24]. The major limitation of their investigation was that it was only tested with one participant. Another study presented a slip and trip detection method using a smart phone placed in the participant’s pocket with data collected in a simulated construction environment [26]. It has shown slip detection accuracy as high as 88%, but neither the types nor sizes of slips detected were discussed. Okita et al. developed an algorithm for slip detection in robots using IMUs [27,28]. Unscented Kalman filter (UKF) was formulated based on a simple dynamic model as a block on a slope without translation. It estimated foot kinematics using IMU measurements. Then, a binary Bayes filter used the error from UKF to estimate the probability of gait and slip states. When tested in a level walking with human subjects, a false negative rate of 35% and a false positive rate of 41% were reported for slip detection [27,28]. A more recent study by Wu et al. developed a deep three-dimensional convolutional neural network to detect slips that occurred on inclined ice surfaces using a GoPro camera [29]. The machine learning model was trained and tested with 360 video clips that consisted of 180 slips and 180 normal steps. The overall slip detection accuracy was 86% with sensitivity and specificity of 81% and 91%, respectively. However, it does not differentiate different types of slips and has small slip sample size. Therefore, the existing findings on slip detection methods are limited and the potential for use with the MAA test remains unclear as there have been limited investigations involving slip detection on inclined surfaces.

### 1.3. Objective

To address this gap, the objective of the present study is to evaluate the ability for a machine learning algorithm trained using data from an optical motion capture system to detect and classify slips occurring during inclined walking on icy surfaces as part of the MAA test.

## 2. Materials and Methods

### 2.1. Data Collection

Nine healthy young adults (6 males, 3 females) were recruited to participate in this study. The demographic information of each participant is listed in Table 1. Their average age, height, and weight were 27 ± 6 years old, 1.76 ± 0.05 m, and 73 ± 10 kg, respectively. To meet the inclusion criteria, they needed to be able to walk up and down slopes ranging from 0° to 15° independently for 45 min. Exclusion criteria included musculoskeletal disorders, cardiopulmonary disorders, orthopedic disease, and any other condition that would impair mobility.

It is difficult to perform sample size calculations for machine learning projects like this one since the sample size depends on various factors, such as complexity of the model, number of input features for the model, strength of the features, etc. We relied on a commonly accepted rule of thumb to collect 10 times more slips per participant than the number of features of interest [2]. Since we planned to extract 36 features to train our classifiers, we would need 360 slips per participant. We chose a relatively small sample (*n* = 9) for this initial proof-of-principle study to determine if these methods were feasible before collecting data from a larger, more representative sample.

Data collection was done in WinterLab, a winter environment simulation laboratory that is located at the KITE Research Institute, Toronto Rehabilitation Institute—University Health Network. The laboratory contains a 4.5 m by 4.6 m ice floor that is approximately 2.5 cm thick. This ice floor was cooled using glycol tubes to 0.5 ± 1.0 °C and the ambient air temperature was maintained at 8.0 ± 2.0 °C for the duration of the data collection sessions. WinterLab was mounted on an electrical screw jack platform capable of tipping and maintaining the entire lab at a slope between 0° and 15° in one-degree increments, as shown in Figure 1 below.

All participants wore a safety harness attached with a fixed line from the upper back to a passive low-friction overhead trolley. Participants were asked to walk up and down a variety of icy slopes while wearing three different models of footwear. The three footwear models were selected to have a range of different MAA scores (based on previous testing) as shown in Table 2. Data were first collected from participants wearing the footwear with the lowest MAA score. Participants switched to the footwear with the next lowest MAA score when they were no longer able walk up the slopes with the first pair without slipping. Participants also walked along a 0° slope (level surface) using all three footwear models. At each slope angle, each participant was asked to perform five walking trials. Each trial consisted of a participant walking up and down the ice-covered walkway at their self-selected pace.

Reflective markers were placed in clusters of three or four markers on the anterior, posterior and lateral aspects of the footwear, shown in Figure 2. A 14-camera passive motion tracking system (Raptor-E, Motion Analysis, Rohnert Park, CA, USA) utilizing Cortex (5.2.0.1518, Motion Analysis, CA, USA) software was used to track the positions of the reflective markers with a sampling frequency of 150 Hz.

### 2.2. Data Analyses

After filling gaps in the motion capture marker trajectories using the Cortex software, the kinematic data were fed into the slip detection algorithm developed in MATLAB (R2014a, MathWorks, Natick, MA, USA), shown in Figure 3 below. They were first filtered in the signal preprocessing stage, then broke down into different signal segments that correspond to different steps. A set of 36 features were extracted from each step. The selected subset of the 36 features were input to both toe slip classifier and heel slip classifier. Both classifiers were 3-class linear Support Vector Machine (SVM) classifiers, but they were trained and optimized for different slips. The toe slip classifier classified the input step as forward toe slip, backward toe slip, or step with no toe slip. The heel slip classifier classified the input step as forward heel slip, backward heel slip, or step with no heel slip.

For overall binary slip detection, the algorithm used the output from both toe slip and heel slip classifier. A slip was detected if the step was classified as either toe slip or heel slip. In other words, a step was only classified as step without slip if it was classified as step with no toe slip and step with no heel slip.

### 2.3. Signal Preprocessing

The marker position data were filtered using a fourth-order, zero-lag, dual-pass Butterworth filter with a 12 Hz cut-off frequency and differentiated to obtain the velocity signals. Finally, a participant-centric frame of reference was applied to all data such that the forward movement of the participants along the anterior-posterior axis was defined to have positive velocity.

### 2.4. Stride Segmentation

O’Connor et al. reported a segmentation algorithm that identified toe off and heel contact using the vertical velocity of the foot [30]. For toe off, it searched for the largest peak in vertical velocity within the order of one typical gait cycle (0.8 s). For heel contact, it searched for series of troughs in the signal using a 0.08-s window. The heel contact was identified as the second largest trough with height below 35% of the range of heel heights encountered during the trial and that occurred between the current stride’s toe off and next heel contact.

Like O’Connor’s algorithm, the stride segmentation algorithm in this study searched for series of peaks and troughs in the vertical velocity of the foot. However, toe off and heel contacts were identified with different sets of constraints and combinations of the following signals, shown in Figure 4 and Figure 5, respectively:Vertical heel marker and toe marker velocities;Angle between foot and floor (foot angle);Angular velocity of the foot.

Details of the toe off and heel contact event identification are described in Section 2.4.1 and Section 2.4.2, respectively. Data that include incomplete strides or movements where the participant was changing direction were discarded.

#### 2.4.1. Toe Off

To identify the toe off events, our algorithm firstly searched for the largest peak in the heel vertical velocity signal with several empirically defined constraints. The threshold for the peaks were 0.13 m/s or greater and they were separated by 90 frames (equivalent to 0.6 s). Toe off was empirically defined as occurring five frames (0.03 s) after the largest peak.

#### 2.4.2. Heel Contact

Figure 6 illustrates the heel contact identification process flow. The criterial for identifying heel contact was empirically determined based on pilot testing.

After toe off and heel contact were identified, these events were used to find the start and end of the complete strides. The start of a complete stride was defined as the midpoint between the toe off of the current stride and the heel contact of the previous stride. The end was defined as the mid-point between the heel contact of the current stride and the toe off of the next stride.

It was important to identify toe off and heel contact accurately, since these events were used to extract many of the features described in the next section. One-hundred non-slip-steps and 100 slip-steps were randomly selected to be evenly distributed among the nine participants. Using the MATLAB random number generator, 11 non-slips-steps and 11 slip-steps were randomly selected from each of the 9 participants. The last 100th non-slip-step and slip-step were randomly selected from any of the participants. The timing of toe off and heel contact identified by the algorithm were compared with those identified manually by viewing the motion capture data frame-by-frame.

### 2.5. Feature Extraction and Selection

A number of features were extracted from the anterior-posterior (AP) velocity and vertical velocity of the heel and the toe, as well as the position signals for each step. Figure 7 compares the velocity signal profile of a normal step to the different types of slips. The arrows in Figure 7 highlights the key differences between slips and normal steps. Typically, the most distinct differences in velocity signals were found before or during toe off for both backward and forward toe slips when comparing to normal steps. For backward and forward heel slips, they occurred after heel contact.

A list of all features and their description are listed in Table 3. A subset of these features, noted in Table 3, were found to be useful and were input into the classifiers in the next step.

Feature 31 and 34 calculated curvature values based on the equation below:(1)$$\kappa \left(t\right)=\frac{{v(t)}^{\prime \prime }}{{\left(1+{{v(t)}^{\prime }}^{2}\right)}^{\frac{3}{2}}}$$
where *κ* represents the curvature at frame *t* and *v* represents the heel AP velocity at frame *t*. Couple time points during a step were also used to compute the features. $T_{heel-off}$ indicates when the velocity of the foot is zero right before heel off in the gait cycle. It is empirically determined that $T_{heel-off}$ occurs when the heel AP acceleration reaches 0.5 m/s^2^ and the heel AP velocity drops below 0.1 m/s. $T_{\max velocity}$ is when the heel AP velocity reaches its maximum value.

Visual inspection of histograms showing the distribution of feature values was used for feature selection. A sample histogram is shown in Figure 8. The *x*-axis is the range of feature values. A bin size of 50 was used to generate the points on the line. The *y*-axis shows the count normalized to the population of the specific class. This histogram shows the distribution of negative AP velocity peaks before toe off for the three classes in the toe slip classifier. Thus, there are three colors, corresponding to each class. The blue line represents the normal step (NS) class, the red line represents the backward toe slip (BTS) class, and the green line represents the forward toe slip (FTS) class. The more overlaps between distributions of feature values, the less useful that feature is for distinguishing the different classes. Features where the overlapping areas between all combinations of the classes were equal to or greater than 75% were discarded.

### 2.6. Slip Classification

Different selected features were normalized and input into the two classifiers in the algorithm. Toe slip classifier was trained to classify toe slips specifically, whereas the heel slip classifier only classified heel slips. Together, the algorithm was able to classify slips as one of four types of slips:Backward toe slips (classify by toe slip classifier);Forward toe slips (classify by toe slip classifier);Backward heel slips (classify by heel slip classifier);Forward heel slips (classify by heel slip classifier).

A backward toe slip was defined as a backward movement of the foot before toe off. Similar to the findings by Powers et al. and Yamaguchi et al., it was characterized by a negative AP velocity of the toe before toe off [31,32]. Forward heel slips were defined by forward movement of the foot after heel contact and was characterized by a positive AP velocity of the heel after heel contact [33]. In some cases, more than one classifier may have identified a slip on the same step, such as backward toe slip and backward heel slip both occurred in one step.

Based on the characteristics of the extracted features, we believed that the data are linearly separable and thus a linear kernel would perform well. In pilot testing, we compared performance of different kernels (linear, polynomial with degree of 2 to 4, and radial basis function kernel). Linear SVM had the best overall performance and least time to train. Further pilot testing was undertaken to compare between multi-class logistic regression, multi-class linear SVM, and artificial neural network (ANN) with one hidden layer and linear SVM was selected based on its performance.

#### 2.6.1. One-versus-Rest Multiclass Classification

One-versus-Rest approach was used to implement the 3-class linear SVM classifiers. In this approach, it splits the multi-class classification into one binary classification per class. For example, one of the binary classifiers for toe slip classifier would be trained to classify a step without toe slip vs. backward and forward toe slip. Thus, there were three binary classifiers for both toe slip and heel slip classifiers. Each individual binary classifier output a probability-like score and the one with the largest score was used for the prediction.

#### 2.6.2. Leave-One-Subject-Out Cross Validation

Leave-one-subject-out cross validation (LOSOCV) was used to measure the performance of both linear SVM classifiers. In this method, one participant was reserved as a test set to provide an unbiased evaluation of the classifier. It tested whether the classifiers could maintain similar performance when encountering a new participant with different gait pattern. The remaining data were divided into a training and validation set at 80:20 ratio. They were used for training each of the linear SVM classifiers while tuning model hyperparameters. This process was repeated so that the data from each participant were used as the test case once.

#### 2.6.3. Handling Imbalanced Data

Since there were significantly more non-slips than slips in the collected data (Table 4), the training and validation data set were imbalanced. This could lower the SVM classifiers performance on detecting slips. Thus, a random under-sampling approach was used to address this issue. In this approach, a subset of training and validation datasets were randomly selected, containing a similar ratio of non-slips and different types of slips. It was repeated 10 times, so that each classifier was trained and tested 10 times to obtain the average performance.

#### 2.6.4. Performance Evaluation Metrics

The classifier performance was evaluated with the following:(2)$$\mathrm{Precision}\;\left(\mathrm{class}=a\right)=\frac{tp\;\left(\mathrm{class}=a\right)}{tp\;\left(\mathrm{class}=a\right)+fp\;\left(\mathrm{class}=a\right)}$$
(3)$$\mathrm{Recall}\;\left(\mathrm{class}=a\right)=\frac{tp\;\left(\mathrm{class}=a\right)}{tp\;\left(\mathrm{class}=a\right)+fn\;\left(\mathrm{class}=a\right)}$$
(4)$$F_{1}\;\mathrm{score}\;\left(\mathrm{class}=a\right)=2*\frac{\mathrm{precision}\;\left(\mathrm{class}=a\right)*\mathrm{recall}\left(\mathrm{class}=a\right)}{\mathrm{precision}\;\left(\mathrm{class}=a\right)+\mathrm{recall}\left(\mathrm{class}=a\right)}$$
where tp represents the true positives, fp represents the false positives, fn represents the false negatives, and a represents a class predicted by the 3-class linear SVM classifier. Each class of the 3-class linear SVM classifier or type of slips would have a precision, recall, and F_1_ score.

#### 2.6.5. Overall Slip Detection

The overall slip detection was merely a logical operation on the output of both toe slip and heel slip classifiers. It was determined based on the condition below where ∨ represents “inclusive or” operation.
(5)Slip=backward toe slip ∨ forward toe slip ∨backward heel slip ∨forward heel slip

The F_1_ score for the overall slip detection performance was also evaluated.

#### 2.6.6. Sensitivity Analysis

Since the extracted features were based on toe off and heel contact, it is important to evaluate whether toe off and heel contact timing detection errors affect the toe slip and heel slip classifier performance. For these selected 200 steps in Section 2.4, their toe off and heel contact timings were varied independently by −15 to +15 frames in intervals of 1 frame. This was to simulate errors in detection. Features were then extracted based on these modified timings and fed into the trained and optimized toe slip and heel slip classifier. The F_1_ scores were then calculated and compared with the F_1_ score without introduction of the errors.

## 3. Results

### 3.1. Types of Slips

A total of approximately 11,000 steps were collected from nine participants. These included approximately 4700 slips distributed across slopes from 0° to 12°, shown in Figure 9. These were categorized into four types of slips (Figure 7): backward toe slips, forward toe slips, backward heel slips, and forward heel slips. The number of each type of slip collected per participant is shown in Table 4. The total number of slips in Table 4 is greater than 4700 because some steps included both toe slips and heel slips.

### 3.2. Stride Segmentation Performance

The error in toe off detection timing was 0.9 ms with the limits of agreement of −24.6 and 26.3 ms (Figure 10). For heel contact detection, the error was 2.4 ms and the limits of agreement were −28.2 and 32.9 ms (Figure 11). The sensitivity analysis showed that errors in toe off detection timing did not affect toe slip classification and heel slip classification and error in heel contact detection did not affect toe slip classification. At the lower limit of agreement of error in heel contact detection, heel slip classification only decreased by 4%.

### 3.3. Slip Detection Performance

The selected features for each of the toe slip and heel slip classifiers are noted in Table 3. As mentioned in Section 2.6, LOSOCV was done to evaluate the accuracy of each classifier model where at each iteration, one subject’s entire data were left out of the training set and used as the test sequence to measure the accuracy of the classifier. The results of LOSOCV for toe slip SVM classifier are shown in Figure 12 and Table 5. The F_1_ scores for non-toe slips and backward toe slips were consistently above 93% for all participants. However, the classification of forward toe slips was poor and fluctuates among participants. The average F_1_ score for the toe slip classifier was 85.7%.

For the heel slip SVM classifier, its LOSOCV results are shown in Figure 13 and Table 6. Its performance was relatively consistent across participants, except for backward heel slip classification of Participant 4. The average F_1_ score for the heel slip classifier was 87.5%.

For overall slip detection performance, the precision, recall, and F_1_ score were 89.1%, 91.5%, and 90.1%, respectively.

## 4. Discussion

### 4.1. Types of Slips

Four types of slips were observed in this study:○Backward toe slips;○Forward toe slips;○Backward heel slips;○Forward heel slips.

Of these, forward heel slips are considered the most hazardous [31,34]. Heel slip occurs in the early stance phase when the weight is transferred to the leading limb. It causes the base of support to move away from the center of mass, causing instability of the body. However, toe slips, which occur in the late stance phase, happen when the weight transfer is almost complete and the body is relatively more stable. It is important to differentiate between the toe and heel slips because this information may be useful for helping footwear manufacturers redesign their footwear appropriately.

Backward toe slips occurred most often when participants ascended the icy sloped walkway. Higher shear forces were generated for push-off in order to overcome the additional work from gravity [35]. In contrast, forward heel slips occurred more frequently when descending. Redfern et al. reported that the shear force near the heel contact phase increases as the ramp angle increases for downhill walking [36]. This increased shear force led to higher probability of forward heel slips. In addition, a variant of forward heel slip, which was unseen in the literature, was identified (Figure 7F). The AP velocity signal was similar to a normal step; it had no local maxima after heel contact but generally had higher AP heel velocity at heel contact. The slip motion blend in with the motion expected from a normal step. This type of slip may occur when participants perceived that they were about to slip forward in a known slippery environment. Thus, they intentionally let the slip to occur, moved with the slip, and then tried to stop the motion gradually slow. Additional studies are needed to investigate further and explain its occurrence.

Redfern et al. also reported rearward motion of the foot upon heel contact, which was classified as backward heel slip in this study. Compared to downhill walking, it was more likely to occur during uphill walking because the leading foot also needed to push back to propel the body up and forward, like lagging foot during toe off.

To the best of author’s knowledge, no study has reported on forward toe slip, which is a forward movement of the foot before toe off. Compared to a normal step, its AP heel velocity has one or more local maxima before toe off. In some cases, these local maxima might not be very distinguishable from the main maxima. Forward toe slip had a rare occurrence of approximately 2% on ice and it always occurred on the lagging foot, following forward heel slip of the leading foot during downhill walking. This may explain why it is not reported in the literature since it may be considered as a consequence of the forward heel slip.

### 4.2. Stride Segmentation Performance

The stride segmentation block of our algorithm was shown to be accurate and robust. The systematic error and the limits of agreement for both toe off and heel contact were less than 1 and 5 frames, respectively. A sensitivity analysis showed that heel contact and toe off timing detection errors did not have an impact on toe slip and heel slip classifier, respectively. This performance was expected because almost all selected features for the toe slip classifier are derived near the toe off event but not the heel contact event (Table 3). We also expected similar performance for the heel slip classifier, where the selected features are derived near the heel contact event. We calculated a 4% decrease seen in heel slip classifier performance when heel contact was detected at the lower limit of agreement. It is not likely meaningful because only a small number of heel contacts is identified at the lower limit of agreement in Figure 11.

### 4.3. Slip Detection Performance

The toe slip classifier average F_1_ scores for non-toe slip (98.0%) and backward toe slip (97.3%) were excellent. Non-toe slips were steps that do not have toe slips; it can be steps with no slips, backward heel slips, or forward heel slips. The overall classifier performance was brought down to 85.7% by poor detection of forward toe slips (54.7%). Compared with other type of slips, there were much less data collected for forward toe slips. Collecting additional data and generating new features may help increase its detection performance. However, this type of slip was considered less important because it was extremely rare and always associated with forward heel slip of the leading foot, which were identified with much better accuracy (Table 6). It is more important to identify forward heel slip that can potentially lead to a forward toe slip. Furthermore, part of the low detection accuracy can be attributed to the errors in the ground truth since human observers had difficulty distinguishing between a forward toe slip and abnormal swing or toe off.

The heel slip classifier performance was good and consistent across nine participants, except for the backward heel slips which have the slip distance of less than 1 cm. The heel slip classifier performance was lower than that of toe slip classifier without the forward toe slip. This may be partly due the forward heel slip variant. Unlike typical forward heel slips, it does not have one or more peaks in AP heel velocity after heel contact. Thus, less useful information could be extracted from the selected features for classification of these slips, resulting in lower classification performance.

The overall slip detection F_1_ score was 90.1%, which was higher than that of both classifiers. Some steps can have both toe slip and heel slip. As long as one of the classifiers correctly predicts a slip occurred, the algorithm would correctly identify it as a slip. For example, a step going uphill can have backward heel slip and backward toe slip. If the output of toe slip and heel slip classifier are backward toe slip and heel slip classifies, respectively, the final output of the algorithm is still a slip. Although the detection for forward toe slip was poor, it did not limit the overall slip detection. With forward toe slips, there was added forward momentum of the body resulting from the forward heel slip of the contralateral foot, described previously. The additional momentum would likely cause a forward heel slip in the same step, which was detected with a much higher accuracy. The limiting factor in the overall accuracy of slip detection was the heel slip classifier performance, which was lower than the toe slip classifier performance when forward toe slip was excluded.

In addition, the algorithm performance would vary for slips with different slip distances. More specifically, the algorithm’s accuracy is expected to increase as slip distance increases. The slip data in this study have slip distances ranging from below 1.5 cm to above 6 cm. For slips with greater slip distances, they tend to be more severe slips with greater changes in the velocity profiles compared to the normal steps. Smaller slips may even be too difficult for the observer to reliably detect; slips smaller than 3 cm are not as likely to result in a gait disturbance [33,37,38,39]. The F_1_ score for overall slip detection increased to 91.0%, if slips with less than 1.5-cm slip distance were excluded.

This study presented the first algorithm that can automatically detect and classify four different types of slips: backward toe slips, forward toe slips, backward heel slips, and forward heel slips. Compared to Wu et al.’s study, our algorithm reported higher slip overall detection performance in F_1_ score (90.1% vs. 79%) [29]. Although our study involved half the number of participants, our algorithm was trained and evaluated with 27 times more numbers of slips. It has a relatively lower chance of overfitting. The advantage of Wu et al.’s algorithm is that it uses GoPro camera video data. Motion capture system can only be set up in in-door lab environment and is more difficult to acquire and set up compared to a video camera. Our algorithm may be more suited for detailed analysis in a lab environment, whereas Wu et al.’s algorithm may be extended to detection in an outdoor environment.

### 4.4. Limitations and Future Work

The main limitation of the present work is that our algorithm was trained and validated on data from only nine young healthy participants. The algorithm should be trained and tested with a larger number of participants and on a wider range of participants to evaluate its generalizability. It is likely that slip detection accuracy will be lower for other populations, such as older adults or mobility-impaired individuals. Future work will include retraining and evaluating the algorithm with more diverse populations, as well as comparing algorithm performance with manual observer for the MAA test. In addition, it will also be important to determine how the performance of our algorithm changes when detecting slips with different severities and how does it compare with a manual observer.

## 5. Conclusions

In this study, we presented an automatic slip detection algorithm with linear multi-class SVM classifiers that was trained and validated on data from approximately 11,000 steps from 9 healthy participants. Our algorithm was able to segment kinematic signals from a motion capture system into series of steps regardless of slips. It was able to detect slips with an overall F_1_ score of 90.1% when using LOSOCV. In addition, the algorithm was able to accurately classify slips as toe slips with F_1_ scores of 97.3% for backward toe slips and 54.5% for forward toe slips, and heel slips with F_1_ scores of 80.9% for backward heel slips and 86.5% for forward heel slips. This study demonstrated promising results for the algorithm with healthy young adults. The proposed algorithm can be applied to detect slips in other lab settings with different floor conditions and footwear styles. Future works can further evaluate and optimize the algorithm with older populations or mobility impaired individuals.

## Figures and Tables

**Figure 1 sensors-22-02370-f001:**
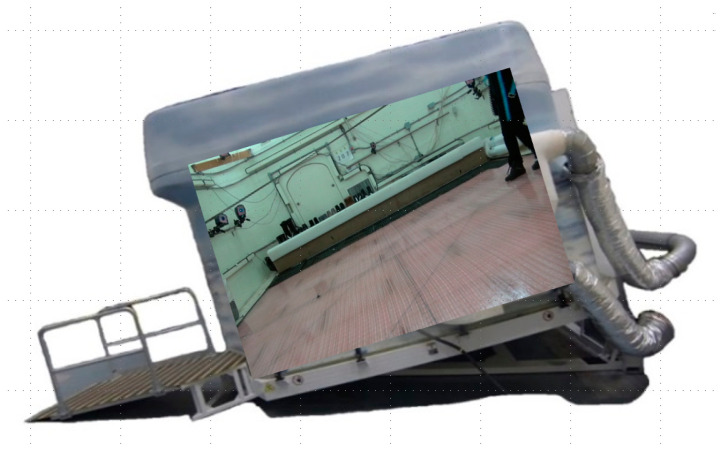
WinterLab shown at a tipped angle.

**Figure 2 sensors-22-02370-f002:**
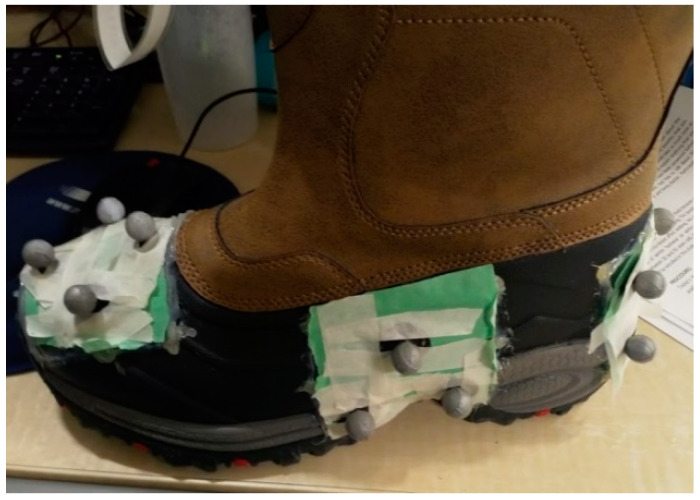
Reflective marker clusters were place on the anterior, posterior and lateral aspects of each pair of test footwear.

**Figure 3 sensors-22-02370-f003:**
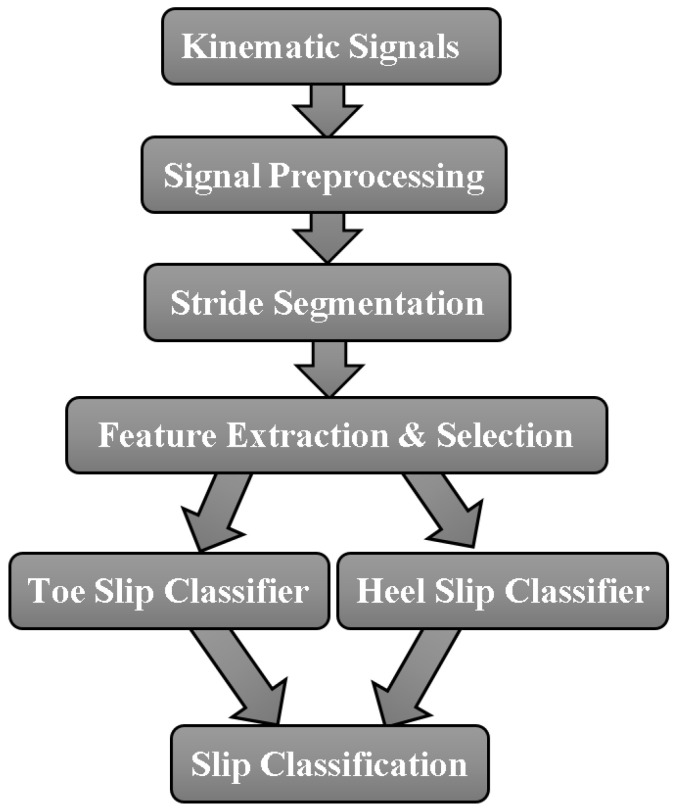
Overview of the slip detection algorithm developed and evaluated in this study.

**Figure 4 sensors-22-02370-f004:**
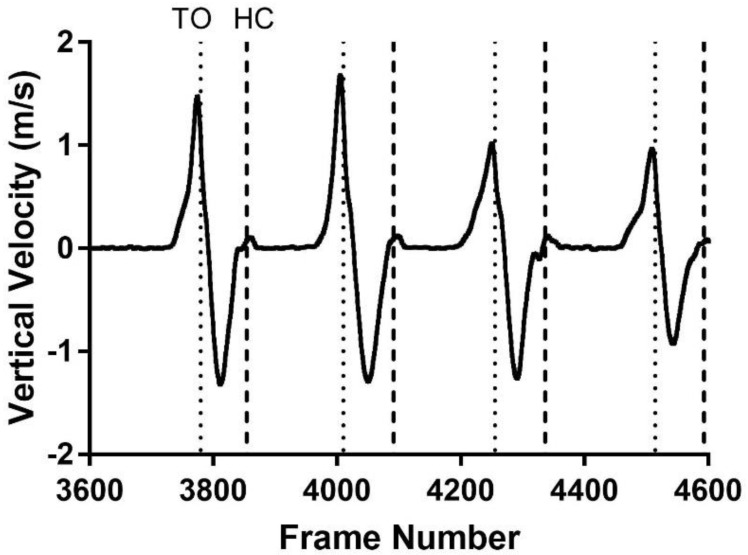
Changes in the vertical heel marker velocity over sample steps with toe off (TO) and heel contact (HC) times shown as dashed vertical lines.

**Figure 5 sensors-22-02370-f005:**
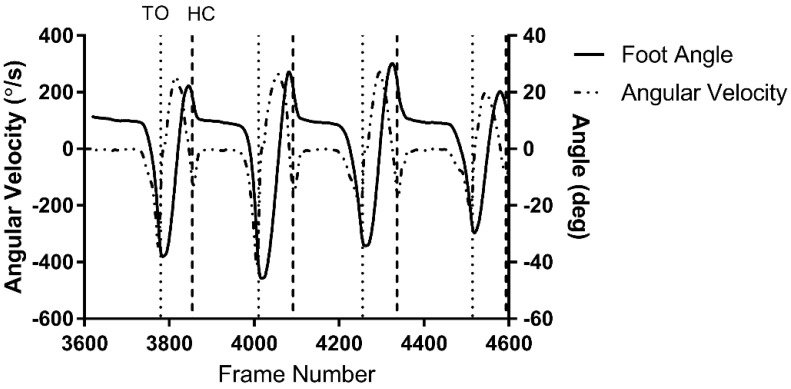
Changes in foot angle and its angular velocity over sample steps with toe off (TO) and heel contact (HC) times shown as dashed vertical lines. The foot angle signal shown does not start at 0°. This offset angle is the result of toe markers and heel marker not being at the same height.

**Figure 6 sensors-22-02370-f006:**
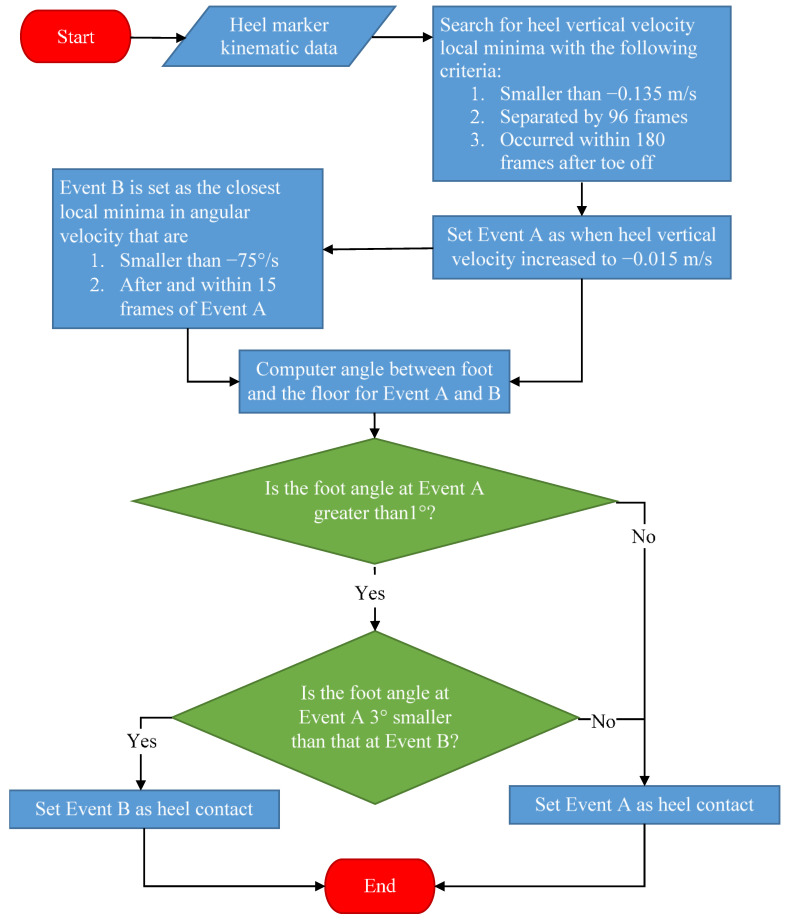
Heel contact event detection process flow chart.

**Figure 7 sensors-22-02370-f007:**
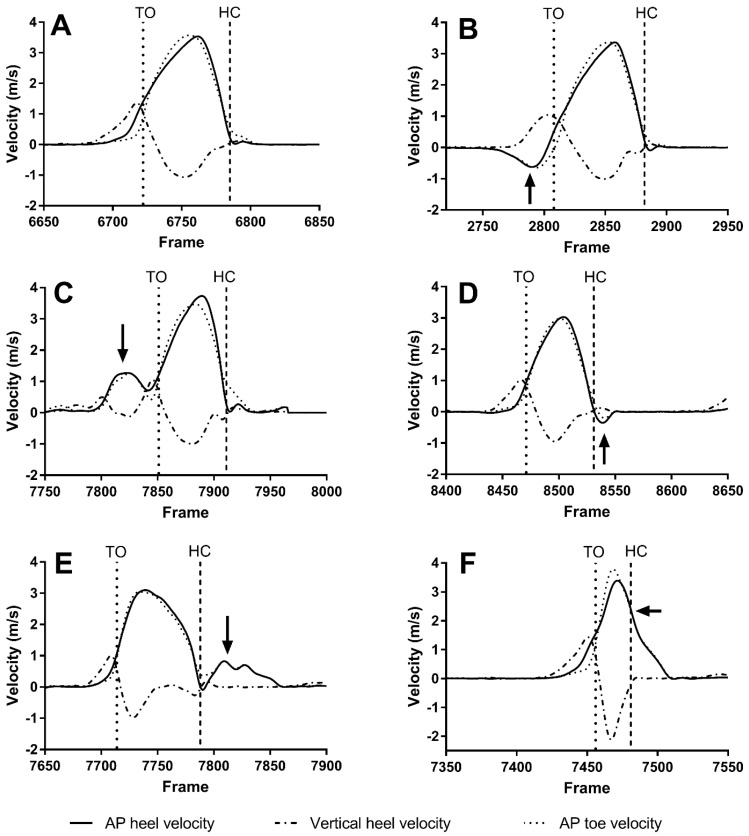
Velocity signal profiles of a normal step and the different types of slips observed. (**A**) Normal step. (**B**) Backward toe slip. (**C**) Forward toe slip. (**D**) Backward heel slip. (**E**) Forward heel slip. (**F**) Forward heel slip variant. The toe off (TO) and heel contact (HC) are shown as dashed vertical lines. The arrows indicate the key features of each slip.

**Figure 8 sensors-22-02370-f008:**
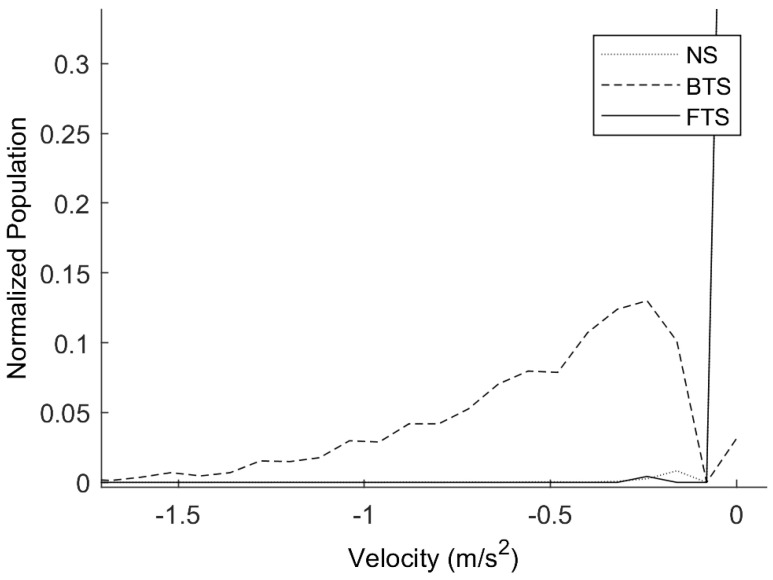
Sample histogram of a feature, negative AP velocity peaks before toe off.

**Figure 9 sensors-22-02370-f009:**
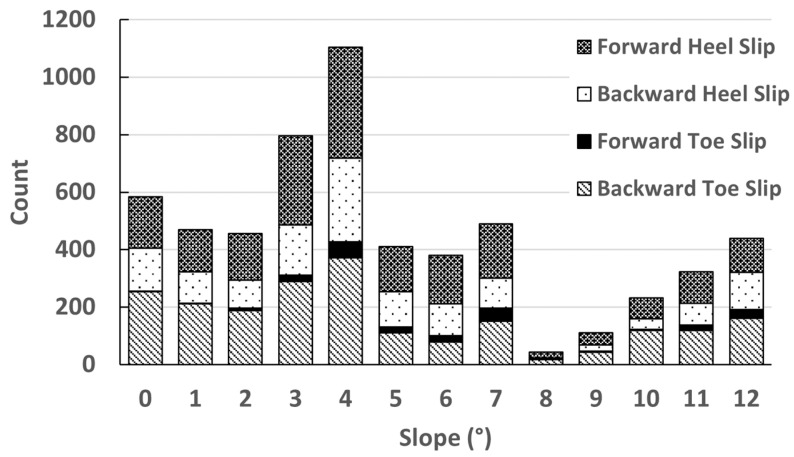
The distribution of different slips recorded across different slopes.

**Figure 10 sensors-22-02370-f010:**
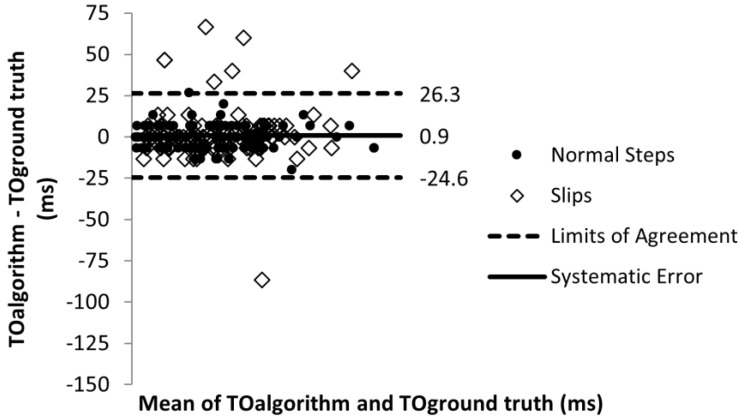
Toe off timing detection error.

**Figure 11 sensors-22-02370-f011:**
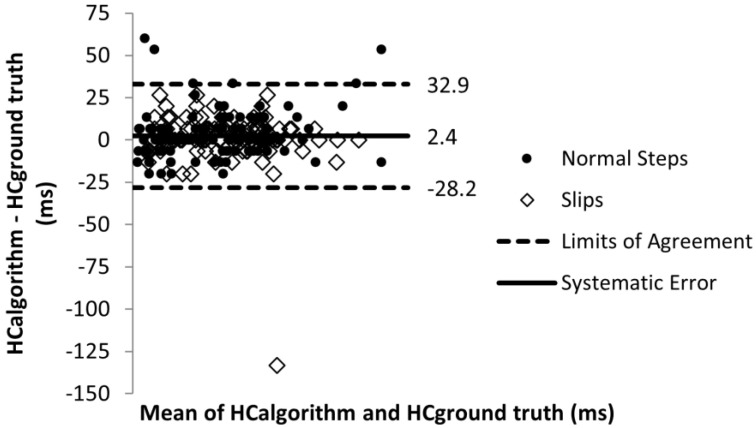
Heel contact timing detection error.

**Figure 12 sensors-22-02370-f012:**
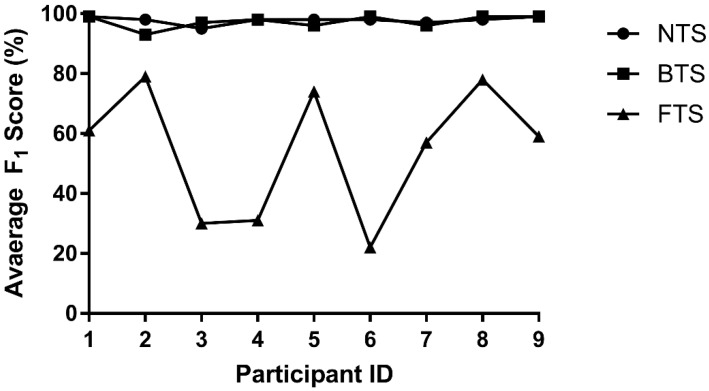
LOSOCV for toe slip classifier. NTS represents non-toe slips, BTS represents backward toe slips, and FTS represents forward toe slips.

**Figure 13 sensors-22-02370-f013:**
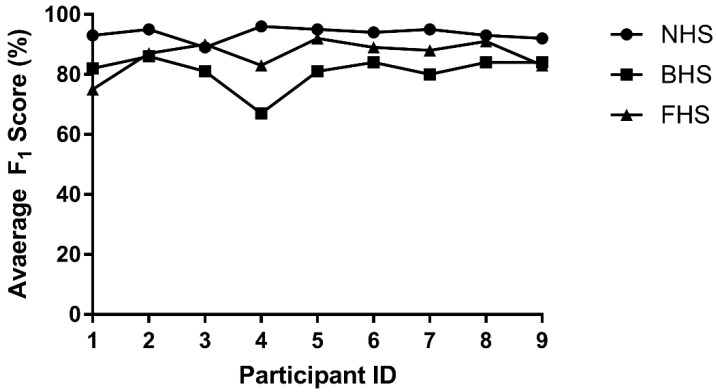
LOSOCV result for the heel slip classifier. NHS represents non-toe slips, BHS represents backward toe slips, and FHS represents forward toe slips.

**Table 1 sensors-22-02370-t001:** Participants’ demographic information.

Participant ID	Gender	Age	Height (cm)	Weight (kg)
1	M	36	177	85
2	M	32	175	49
3	M	20	170	73
4	F	21	179	73
5	F	34	168	68
6	M	29	175	73
7	M	22	183	80
8	F	23	179	84
9	M	24	175	75

**Table 2 sensors-22-02370-t002:** Footwear models used for data collection.

Footwear Model	MAA Score	Range of Slopes Covered
Canadian Tire Woods Snow Peak Boots (1871132)	0°	0° to 4°
Mark’s WindRiver Canmore (5CPEWRF16-5224)	4°	0°, 3° to 7°
Mark’s WindRiver Mallory (5DQEWRFW5134)	11°	0°, 8° to 11°

**Table 3 sensors-22-02370-t003:** List of features for each classifier.

Feature Number	Feature Description
1	Number of negative peaks separated by 100 ms in heel vertical velocity
2	Number of positive peaks separated by 100 ms in heel vertical velocity
3 ^1^	Number of positive peaks separated by 7 ms in toe vertical velocity between toe off and heel contact
4 ^1^	Difference of the number of positive peaks and negative peaks separated by 7 ms in toe vertical velocity between toe off and heel contact
5 ^1,2^	Heel AP velocity at heel contact
6	Time it takes heel AP velocity to reach zero after heel contact
7 ^2^	Area of the heel AP velocity from heel contact to the point where velocity reaches zero
8 ^2^	Area of the negative peak in heel AP velocity immediately after heel contact
9 ^2^	Area of the negative peak in heel AP velocity after heel contact that is different from the peak in Feature 8
10 ^1,2^	AP displacement of the heel between heel contact and mid-stance
11 ^1,2^	Number of positive peaks in heel AP velocity after heel contact
12 ^1,2^	Velocity of the largest positive peak in heel AP velocity after heel contact
13 ^1,2^	Area of positive peaks in heel AP velocity after heel contact
14 ^1,2^	Area of positive peaks in heel AP and medial-lateral velocity after heel contact
15 ^1,2^	Sum of Feature 7 and Feature 10
16 ^1,2^	Difference between Feature 13 and Feature 8
17 ^1,2^	Binary feature that describes whether the foot comes to a full stop after heel contact. The criteria for full stop are that its absolute acceleration needs to be smaller than 0.5 m/s^2^ and its absolute velocity needs to be smaller than 0.01 m/s.
18 ^1^	Number of positive peaks in heel vertical velocity before toe off
19 ^1^	Number of positive peaks in toe vertical velocity before toe off
20 ^1^	Number of positive peaks in heel AP velocity before toe off
21 ^1^	Number of positive peaks in toe AP velocity before toe off
22 ^1^	Maximum velocity of the largest positive peak in heel AP velocity before toe off
23 ^1^	Maximum velocity of the largest positive peak in toe AP velocity before toe off
24 ^1,2^	Heel AP velocity at $T_{heel-off}$
25 ^2^	Toe AP velocity at $T_{heel-off}$
26 ^1^	Number of negative peaks in toe AP velocity before toe off
27 ^1^	Width of the largest negative peak in toe AP velocity before toe off
28 ^1^	Maximum velocity of the largest negative peak in toe AP velocity before toe off
29 ^1^	Area of all negative peaks in toe AP velocity before toe off
30 ^2^	Number of positive peaks in heel AP velocity after $T_{\max velocity}$
31	Sum of the curvature values between $T_{heel-off}$ and $T_{\max velocity}$ $\overset{T_{\max velocity}}{\underset{t=T_{heel-off}}{\sum }}\kappa \left(t\right)$
32	Mean of the curvature values between $T_{heel-off}$ and $T_{\max velocity}$ $\frac{1}{T_{\max velocity}-T_{heel-off}\;}\overset{T_{\max velocity}}{\underset{t=T_{heel-off}}{\sum }}\kappa \left(t\right)$
33 ^1,2^	Sum of the curvature values between $T_{\max velocity}$ and $T_{heel-off}$ of the next step $\overset{T_{heel-off\;of\;next\;step}}{\underset{t=T_{\max velocity}}{\sum }}\kappa \left(t\right)$
34 ^1,2^	Mean of the curvature values between $T_{\max velocity}$ and $T_{heel-off}$ of the next step $\frac{1}{\;T_{heel-off\;of\;next\;step}-T_{\max velocity}\;}\overset{\;T_{heel-off\;of\;next\;step}}{\underset{t=T_{\max velocity}}{\sum }}\kappa \left(t\right)$
35 ^1^	Area between the heel AP velocity curve and a straight line drawn from the point before $T_{\max velocity}$ where the heel AP velocity is zero to $T_{\max velocity}$
36 ^2^	Area between the heel AP velocity curve and a straight line drawn from $T_{\max velocity}$ to the point after $T_{\max velocity}$ where the heel AP velocity is zero

^1^ Feature selected for toe slip classifier. ^2^ Feature selected for heel slip classifier.

**Table 4 sensors-22-02370-t004:** Number of slips collected by type and participant, identified through visual inspection.

Participant ID	Backward Toe Slip	Forward Toe Slip	Backward Heel Slip	Forward Heel Slip
1	210	3	55	224
2	173	46	139	160
3	289	36	255	370
4	114	7	100	110
5	293	30	184	210
6	305	6	168	279
7	245	33	114	258
8	317	60	191	249
9	173	11	226	194
Total	2119	232	1432	2054

**Table 5 sensors-22-02370-t005:** Toe slip classifier performance.

	Non-Toe Slip	Backward Toe Slip	Forward Toe Slip	Average
Precision	99.2%	95.9%	45.2%	80.1%
Recall	96.7%	99.0%	82.2%	92.6%
F_1_ score	98.0%	97.3%	54.7%	85.7%

**Table 6 sensors-22-02370-t006:** Heel slip classifier performance.

	Non-Heel Slip	Backward Heel Slip	Forward Heel Slip	Average
Precision	93.7%	77.4%	90.6%	87.2%
Recall	93.6%	86.4%	83.8%	87.9%
F1 score	93.6%	80.9%	86.5%	87.5%

## Data Availability

The data are not publicly available due to privacy limitation.

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
