# Peer review of "Development and Evaluation of a Slip Detection Algorithm for Walking on Level and Inclined Ice Surfaces"

_sensors, 2022, doi:10.3390/s22062370_

Round 1
Reviewer 1 Report
The manuscript is an applied research work related with the use of an optical motion capture system for the development of an automated slip detection algorithm for walking on level and inclined ice surfaces used with the human-centered Maximum Achievable Angle (MAA) test without the need of human observer.
MAA test is a test developed by the authors team at the KITE Research Institute to identify differences in slip resistance performance. It involves participants walking up and down ice-covered inclines while wearing test footwear in a simulated winter environment but it need a human observer to determine when a participant has experienced a slip. The hypothesis of the authors is that the addition of an automated slip detection system would make the MAA test more objective and accurate.
General comments: As it stands, this paper is difficult to understand because the methodological process is described in a confusing way as well as the classifiers. For example, (i) it is not clear in the text if toe slip classifier is binary and classifies “clip in Backward” versus “forward” or other things (same way for heel slip), (ii) Why did you choose SVM approach, whereas other machine learning algorithms could be appropriate and their performance could be compare? The results are not sufficiently described and the discussion should be expanded. The paper needs to be better explained and justified on many points. In all this makes for a substantial revision request.

Reviewer 2 Report
The paper presents an interesting idea and the outcome of the study will provide benefit for human needs. However, there are a few suggestions for the paper improvement:
- The study uses markers and camera passive motion for gait analysis based on the machine learning method. The slip sensors probably have been presented in previous studies for the inclined surface. Please include these previous studies in the Introduction.
- There are 9 young adults were participated in the study. As slip is also an effect of a mass, I would suggest the Authors include brief information on the participants such as weight and height.
- Table 1 presents the footwear models used for data collection. I suggest including information related to the roughness value of the inclined surface. Because, for example, concrete floor and woods floor has different surface roughness that will affect the slip as well.
- Figures 4 and 5 have very small font sizes on the x- and y-label. It is not readable. Please revise.
- There are dashed and solid lines in Figure 7, what are they? please provide a legend of the Figure.
- There are about 36 features used in the study that was presented in Table 2. It would be better if the Authors provide the Equation for some features.
- Figure 8 is not a standard of figure presentation in an academic paper. Please revise it by removing a grey background.
- The font sizes of x- and y-label of Figures 9, 12, and 13 are also not readable. It is too small.
- In my opinion, more sentences can be added in the Conclusion to highlight the results.
Round 2
Reviewer 1 Report
The new version of the paper presented by the authors has been extended with many explanations and requested complements. I now recommend its acceptance.
Reviewer 2 Report
Dear Authors,
Thank you for providing the revised version. I have checked the documents and I have no further comments.